# High-Intensity Exercise Promotes Deleterious Cardiovascular Remodeling in a High-Cardiovascular-Risk Model: A Role for Oxidative Stress

**DOI:** 10.3390/antiox12071462

**Published:** 2023-07-20

**Authors:** Aline Meza-Ramos, Anna Alcarraz, Marta Lazo-Rodriguez, Gemma Sangüesa, Elisenda Banon-Maneus, Jordi Rovira, Maria Jose Ramirez-Bajo, Marta Sitges, Lluís Mont, Pedro Ventura-Aguiar, Montserrat Batlle, Eduard Guasch

**Affiliations:** 1Institut d’Investigacions Biomèdiques August Pi i Sunyer (IDIBAPS), 08036 Barcelona, Spain; cmeza@recerca.clinic.cat (A.M.-R.); aalcarraz@recerca.clinic.cat (A.A.); mlazo@recerca.clinic.cat (M.L.-R.); gsanguesa@umanresa.cat (G.S.); ebanon@recerca.clinic.cat (E.B.-M.); jrovira1@recerca.clinic.cat (J.R.); mramire1@recerca.clinic.cat (M.J.R.-B.); msitges@clinic.cat (M.S.); lmont@clinic.cat (L.M.); pventura@clinic.cat (P.V.-A.); 2Medicine Department, Universitat de Barcelona, 08036 Barcelona, Spain; 3Consejo Nacional de Ciencia y Tecnología (CONACyT), Ciudad de México 03940, Mexico; 4Laboratori Experimental de Nefrologia i Trasplantament (LENIT), 08036 Barcelona, Spain; 5Cardiovascular Institute, Clínic Barcelona, 08036 Barcelona, Spain; 6Centro de Investigación Biomédica en Red Enfermedades Cardiovasculares (CIBERCV), 28029 Madrid, Spain; 7Department of Nephrology and Kidney Transplantation, Clínic Barcelona, 08036 Barcelona, Spain

**Keywords:** strenuous exercise, high cardiovascular risk, cardiovascular remodeling, kidney disease, metabolic syndrome, animal model

## Abstract

Although the benefits of moderate exercise in patients at high cardiovascular risk are well established, the effects of strenuous exercise remain unknown. We aimed to study the impact of strenuous exercise in a very high cardiovascular risk model. Nephrectomized aged Zucker obese rats were trained at a moderate (MOD) or high (INT) intensity or were kept sedentary (SED) for 10 weeks. Subsequently, echocardiography and ex vivo vascular reactivity assays were performed, and blood, aortas, perivascular adipose tissue (PVAT), and left ventricles (LVs) were harvested. An improved risk profile consisting of decreased body weight and improved response to a glucose tolerance test was noted in the trained groups. Vascular reactivity experiments in the descending thoracic aorta demonstrated increased endothelial NO release in the MOD group but not in the INT group, compared with SED; the free radical scavenger TEMPOL improved endothelial function in INT rats to a similar level as MOD. An imbalance in the expression of oxidative stress-related genes toward a pro-oxidant environment was observed in the PVAT of INT rats. In the heart, INT training promoted eccentric hypertrophy and a mild reduction in ejection fraction. Obesity was associated with LV fibrosis and a transition toward β-myosin heavy chain and the N2Ba titin isoform. Exercise reverted the myosin imbalance, but only MOD reduced the predominance of the N2Ba titin isoform. In conclusion, moderate exercise yields the most intense cardiovascular benefits in a high-cardiovascular-risk animal model, while intense training partially reverts them.

## 1. Introduction

Cardiovascular (CV) diseases are a major public health concern. Their management and, particularly, prevention are hindered by the diversity of risk factors eventually culminating in CV conditions. Obesity, a major risk factor for the leading causes of death, including diabetes mellitus, affects 13% of the world’s population and its prevalence has tripled since 1975, resulting in its classification as a pandemic. Cellular senescence disrupts vascular physiology, promotes atherosclerosis, and mediates, at least partially, the aging-related threat to CV health. Severe renal failure plays an important role in coronary calcification and ischemic heart disease pathology, but even a mild reduction in glomerular filtration rate is associated with an increased CV risk [1]. Interestingly, physical activity may tackle several of these risk factors and help reduce the CV disease burden. Regular moderate physical activity has been demonstrated to reduce incident coronary heart disease and type 2 diabetes [2], while insufficient physical activity increases the risk of all-cause mortality and is strongly associated with a higher risk of presenting with CV diseases [2]. The benefits of regular exercise are mediated at least partially by an enhanced systemic antioxidant capacity [3]. The progressive increase in sedentary lifestyles has prompted international scientific societies to encourage the regular practice of moderate to intense exercise. However, some authors claim that the benefits produced by exercise follow a linear dose–response relationship, with larger benefits reached at higher exercise loads [4].

These data contrast with emerging translational and clinical evidence showing that long-term, high-intensity endurance exercise is associated with a heightened risk of atrial and ventricular arrhythmias [5] and might also enhance coronary calcification [6,7,8] and increase vascular stiffness through repetitive tunica media damage [9]. In a large UK women cohort, the best long-term CV outcomes were achieved with strenuous exercise two or three times per week, while daily strenuous training was associated with an increased risk of stroke and myocardial infarction [10]. An hemodynamic overload [11], along with transient systemic [12] and local [13] inflammatory and oxidative stress [14] cascades have been hypothesized to contribute to these deleterious effects. Overall, these results support the U-shaped relationship between exercise load and health outcomes, which is thought to be driven by the balance between the well-acknowledged benefits and the potentially deleterious effects of physical activity [15]. Under this assumption, individuals at very high CV risk would show no deleterious effects of strenuous exercise, as protection against risk-factor progression would largely overcome any exercise-induced damage. Notably, the low levels of physical activity usually attained by individuals at very high risk impede obtaining conclusive data in humans, and to date, this hypothesis remains unproven.

The objective of our study was to analyze the vascular (elastic arteries) and cardiac (left ventricle) remodeling resulting from strenuous training compared to moderate exercise in a high-CV-risk animal model recapitulating the male sex, aging, obesity, metabolic syndrome, and kidney disease.

## 2. Materials and Methods

All analyses were performed blind to group assignment.

### 2.1. Animal Model and Exercise Protocol

Male Zucker obese rats (200–220 gr Crl:ZUC(Orl)-Leprfa strain 691, Charles River Laboratories, Écully, France), a well-established model of metabolic syndrome, were utilized in accordance with the directives 86/609/EEC and 2010/63/UE under a protocol approved by the Ethics Committee on animal experimentation of the Universitat de Barcelona and Generalitat de Catalunya (approval reference 11273). Animals were housed in a controlled environment at constant humidity and temperature, with a light and dark cycle of 12 h, rodent chow, and tap water ad libitum.

A mild, subclinical reduction in glomerular filtration rate was induced by means of a left nephrectomy. Briefly, 7-week-old rats were anesthetized with inhaled isoflurane (2%), and a small incision was made in the left flank to access the left kidney. The vascular renal hilum was ligated and sectioned, and the left kidney was removed. The incision was subsequently closed with nonabsorbable surgical sutures.

Later, animals were randomly assigned to either remain sedentary (SED, n = 10) or to perform moderate- (MOD, n = 11; 15 cm/s 45 min) or high-intensity exercise (INT, n = 12; 25 cm/s 60 min) on a treadmill. Exercise loads were chosen to fit the physical capacity of these rats and estimated from previous studies on similar animal models to correspond to approximately 50–60% and 85% of maximum capacity for the MOD and INT groups, respectively [11,12]. Before reaching the full exercise protocol, animals were acclimatized to the treadmill for 2 weeks. Speed and time were subsequently and progressively increased until their group objective was reached. Exercise training started at 26 weeks of age and lasted 10 weeks; therefore, rats were analyzed at 36 weeks old. All training sessions were overseen by an experienced investigator to ensure adequate running, avoid harm while training, and prevent psychological or physical stress.

To identify the CV damage induced by the various risk factors in obese (SED, MOD, and INT) Zucker rats, we used a control (CTL) group. The CTL group consisted of sedentary, young (8- to 10-week-old) Zucker lean rats (CR-Zucker LE, Charles River Laboratories, Écully, France) who are considered low-CV-risk, in contrast to the aged, obese rats used in the former model.

### 2.2. Urine and Plasma Collection and Analysis

At baseline (three weeks before training) and 3, 7, and 10 weeks after beginning the training protocol, each rat was housed in a metabolic cage for 24 h, where food and water intake and urinary volume were quantified. Urine was collected, and proteinuria and creatinine levels were determined at the Central Laboratory of the Clínic Barcelona.

Every two weeks and not less than 24 h after the previous training session, blood was obtained through a jugular vein puncture in anesthetized rats. Blood urea nitrogen, creatinine, Na^+^, K^+^, Ca^2+^, Cl^−^, glucose, lactate, hemoglobin concentration, hematocrit, and a blood gas test (pH, pO_2_, pCO_2_, and HCO_3−_) were determined using the EPOC blood analysis system (Siemens Healthcare, Erlangen, Germany) following the manufacturer’s instructions.

Vascular injury biomarkers and cytokine levels were assessed in plasma samples obtained before euthanasia. Blood samples were centrifuged (2000× *g* for 10 min at 4 °C), and plasma was stored at −80 °C until analysis. Later, adiponectin, sE-Selectin, sICAM-1, TNF-α, IL-6, and IL-10 were quantified in plasma samples using two multiplex kits according to the manufacturer’s protocol (MILLIPLEX kits MAP Rat Vascular Injury Magnetic Bead Panel 2—Toxicity Multiplex Assay RV2MAG-26K Millipore; and Rat Cytokine/Chemokine Magnetic Bead Panel—Immunology Multiplex Assay, RECYTMAG-65K Millipore).

### 2.3. Glucose Tolerance Test and Insulin Determination

Before euthanasia, rats were fasted for 12 h, and an intraperitoneal glucose tolerance test (IPGTT) was performed. A total of 2 g glucose per kg of body weight was administered to each rat, and the tail-blood glucose was measured using an automatic glucose monitoring device (Accu-Chek Sensor, Roche) at baseline and 15, 30, 60, 90, and 120 min after glucose administration. Serum samples were also collected before and 30 min after glucose administration for insulin quantification. Insulin was determined in serum samples with an ultrasensitive rat insulin ELISA kit (Mercodia, Uppsala, Sweden).

### 2.4. Echocardiogram

Transthoracic echocardiograms were performed on all rats under inhaled anesthesia (1.5% isoflurane) during the last week of the exercise protocol and at least 12 h after the previous training session. The study was performed with a Vivid q system (GE Healthcare Ultrasound) using a lineal array transducer i12L-RS (5.0–13 Megahertz). A parasternal long-axis view was acquired and processed offline. The M-mode cursor was placed below the level of the mitral leaflet tips, and the interventricular septum (IVS), left ventricle diameter (LVDd), and left ventricle posterior wall (LVPW) were measured at end-diastole; the LV diameter was also measured at end-systole (LVDs). In the same view, the M-mode cursor was set above the aortic valve, and the aortic and left atrial diameters were measured in systole. Measurements were indexed by tibia length. The left ventricular volume (*LVV*) was calculated with the Teichholz formula:(1)$$LVV=\left(\frac{7}{2.4+D}\right)\cdot D^{3}$$
where *D* is the left ventricle diameter either at the end-systole (LVDs) or end-diastole (LVDd). Subsequently, the *LVV* at the end-systole (*LVVs*) and the end-diastole (*LVVd*) were calculated. The left ventricle ejection fraction (*LVEF*) was then quantified as:(2)$$LVEF=\left(\frac{LVVd-LVVs}{LVVd}\right)\cdot 100$$

### 2.5. Euthanasia

Two days after the last training session, animals were euthanized with an isoflurane overdose (inhaled 5%). The thoracic aorta was excised and processed as explained later. The heart was removed, cleaned from blood remains, and weighed. Then, it was sectioned transversally at the mid-papillary level; the basal block was preserved in formol for histological analysis, and the apical block was dissected into the left ventricle (LV), right ventricle (RV), and interventricular septum (IVS) and immediately snap-frozen in liquid nitrogen for later molecular analysis.

### 2.6. Vascular Reactivity

The thoracic aorta was excised in a single segment and maintained in a chilled Krebs solution (NaCl 119 mM, KCl 4.7 mM, KH_2_PO_4_ 1.18 mM, MgSO_4_ 1.17 mM, NaHCO_3_ 24.9 mM, EDTA 0.023 mM, CaCl_2_ 1.6 mM, and glucose 6 mM). Perivascular adipose tissue (PVAT) was dissected and frozen in liquid nitrogen. One aortic ring was preserved in formol, and a second ring was frozen. The remaining descending thoracic aorta was dissected from perivascular tissue and cut into three 3 mm long aortic rings that were mounted on stainless steel hooks under a resting tension of 1.5 g in an organ bath (20 mL Krebs at 37 °C, gassed with a mixture of 95% O_2_ and 5% CO_2_).

All aortic rings were stabilized for 45 min, activated with KCl, and washed; subsequently, the maximum contraction force was obtained and recorded after incubation with 80 mM KCl. The endothelium viability was confirmed by inducing relaxation with carbachol (CCH, C4382 Sigma-Aldrich, St. Louis, MO, USA) 10^−5^ M after contraction with phenylephrine (PHE, P6126 Sigma-Aldrich) 10^−6^ M. Thereafter, cumulative dose–response curves (10^−10^ M to 10^−5^ M) to CCH, PHE, and sodium nitroprusside (SNP, 71778 Sigma-Aldrich) were generated for each ring. One ring was also incubated with a nitric oxide synthase (NOS) inhibitor (LNMMA, M7033 Sigma-Aldrich), a second ring was incubated with a free radical scavenger (TEMPOL, 176141 Sigma-Aldrich), and no additional reagents were used for the third ring.

### 2.7. Heart and Vascular Histology Assessment

#### 2.7.1. Picrosirius Red

The histological pieces of the heart and thoracic aorta kept in formol were later embedded in paraffin and sliced into 4-µm-thick sections. Picrosirius red staining was performed to assess the collagen content. A single microphotograph was obtained in each tissue, scanning the full heart or aortic section (Pannoramic Desk Scan, 3DHistech, Budapest, Hungary). In the heart, myocardial fibrosis burden was measured separately in each wall (LV, IVS, and RV) after excluding endocardial, epicardial, and perivascular fibrosis. In aortic sections, the fibrosis content was evaluated in the tunica media. ImageJ (NIH, Bethesda, MD, USA) was used for semiautomatic quantification. Values of fibrosis are expressed as the percentage of red-stained fibers over the total area.

#### 2.7.2. Aortic NRF2 Immunofluorescence

The presence of the nuclear factor erythroid 2-related factor 2 (NRF2) protein in the nucleus was assessed in 4-µm-thick descending thoracic aorta sections to estimate the activation of antioxidant proteins. Deparaffinization was accomplished with a battery of xylene and decreasing concentrations of ethanol (100% to 70%). The sections were incubated with 10 mM sodium citrate at 80 °C for 40 min for antigen retrieval. A PBS solution with Triton 0.5% (T8787 Sigma-Aldrich), bovine serum albumin 2 m/v% (A3059 Sigma-Aldrich), glycine 0.3 M (G8898 Sigma-Aldrich), and goat serum 5% (G9023 Sigma-Aldrich) was used to block nonspecific binding sites for 60 min. Samples were incubated overnight at 4 °C with an NRF2 primary antibody (Ab76026 Abcam, Cambridge, UK; 1/50 dilution). The next morning, samples were incubated at room temperature with a fluorescent secondary antibody, Alexa Fluor 647 (A32733 Thermo Fisher, Waltham, MA, USA; 1/200 dilution), for 120 min. Then, sections were stained with DAPI (D9542 Sigma-Aldrich) and mounted with Fluoromount (F4680 Sigma-Aldrich). In each sample, three microphotographs were taken in both bandpass filters (365–445 and 640–690). Total and NRF2-positive nuclei were counted, and their ratio was calculated.

#### 2.7.3. Heart Triple Immunofluorescence

Paraffin-embedded LV sections were treated as described in the previous section for deparaffinization, antigen retrieval, and nonspecific binding blockade. Subsequently, samples were incubated overnight at 4 °C with a vimentin primary antibody (Ab92547 Abcam, Cambridge, UK; 1/150 dilution) and the next morning with the secondary antibody goat antirabbit IgG conjugated to Alexa Fluor 405 (A31556 Thermo Fisher, Waltham, MA, USA; 1/200 dilution) at room temperature for 120 min. Additionally, samples were labeled with a wheat germ agglutinin (WGA) lectin conjugated to Alexa Fluor 594 (W11262 Thermo Fisher; 1/200 dilution) and with isolectin GS-IB4 conjugated to Alexa Fluor 488 (I21411 Thermo Fisher; 1/200 dilution) for 120 min. Sections were mounted with Fluoromount (Sigma-Aldrich, St. Louis, MO, USA). Microphotographs were obtained in three different regions at the three different bandpass filters (345–445, 470–525, and 550–605) for both LV and IVS at a magnification of 63×. Myocyte count, cross-sectional area, extracellular matrix content (ECM), capillary, and fibroblast count were assessed with morphological measurements with ImageJ [13].

### 2.8. Expression of mRNA in Cardiac Tissue, Aorta, and PVAT

Total RNA was extracted from frozen 1 cm long sections of the descending thoracic aorta, 100 mg pieces of lyophilized PVAT, and 20 mg pieces of cardiac tissue using TRIzol (Invitrogen Corporation, Carlsbad, CA, USA) for homogenization. Afterward, a mirVana miRNA Isolation kit (Invitrogen) was employed according to the manufacturer’s protocol. RNA integrity and loading amounts were evaluated by analyzing absorbance at 260 nm and 280 nm wavelengths in a Nanodrop 2000 (Thermo Scientific, Waltham, MA, USA). Reverse transcription with random primers was then performed with 1 µg of total RNA, with the aid of a High-Capacity cDNA Reverse Transcription Kit (Applied Biosystems, Waltham, MA, USA) and adding RNase inhibitors (Applied Biosystems) in the MyCycler Thermal Cycler (Bio-Rad Laboratories, Hercules, CA, USA). Gene expression was evaluated utilizing Viia7 Real-Time PCR (Applied Biosystems) with SYBR Green PCR Master Mix reagent (Applied Biosystem). Every PCR was performed in duplicate and normalized to selected housekeeping genes (i.e., Ppib, Hprt, Gusb, Gapdh, and Actb). Specifically, the geometric mean of the two most stable housekeeping genes, as determined with RefFinder (https://www.blooge.cn/RefFinder/), was used for normalization, and the results are shown relative to the sedentary group with the 2^−∆∆Ct^ method. The analyzed genes and specific primers are detailed in Appendix A.

### 2.9. Statistical Analysis

Descriptive statistics are summarized as the mean ± standard deviation (SD) unless otherwise reported. For analyses with a single independent variable, data were compared by one-way (including group as the main factor) ANOVA. In the presence of two independent predictor variables and repeated measurements, data were modeled in a mixed effects model with a repeated measurements covariance structure. Specific details on each model are provided in the figure legends. When the omnibus test showed an overall difference, post hoc comparisons were performed with a least significant difference (LSD, three or fewer post hoc comparisons, which is equivalent to no family-wise error rate correction) test or by computing the false discovery rate (FDR, more than three post hoc comparisons). The normality of the residuals of ANOVA analyses and linear mixed-effects models was assessed in QQplots; when needed, analyses were performed with log-transformed values of the dependent variable. Plotting residuals against the fitted values confirmed homoscedasticity. For vascular reactivity analyses, data were fitted into a three-parameter nonlinear regression model to obtain the maximum effect (Emax) of the agonist and the concentration that elicited half (EC50) of the Emax. Both parameters were compared between groups as previously described for the other variables. A *p*-value of <0.05 was considered to indicate statistical significance in all analyses. Data were analyzed with GraphPad Prism (v8.4, GraphPad Software Inc., Boston, MA, USA) and R (v. 3.6, R Foundation for Statistical Computing, Vienna, Austria) and the *lmertest*, *car*, and *emmeans* packages.

## 3. Results

### 3.1. Exercise Improves Glycosidic Metabolic Profile in Zucker Rats

Weekly measurements of body weight are plotted in Figure 1A. From the 7th training week, the INT group showed a significantly lower body weight than the SED group. At sacrifice, the indexed body weight was progressively lower from SED to MOD to INT (*p* for trend 0.04, Figure 1B). Baseline glycemia was similar in all groups, but blood glucose levels dramatically rose after intraperitoneal glucose overload in SED rats. Both moderate and intensive training similarly blunted the increase in maximal glucose levels (Figure 1C) and the glycemia area under the curve during the IPGTT (Figure 1D). Interestingly, the improved glycemic metabolism in trained rats was associated with higher glycemia-induced insulin release, likely due to the impaired insulin release in sedentary rats (Figure 1E). No changes in plasma triglyceride levels were observed between sedentary and exercised Zucker rats (Figure 1F). Overall, these results highlight a high-CV-risk profile in sedentary Zucker rats that improved in both the moderate and intense training groups.

### 3.2. Unilateral Nephrectomy Elicits a Mild Impairment in Renal Function

In rats from all groups, blood creatinine levels nonsignificantly increased over time (*p* = 0.072, Appendix A). BUN levels fluctuated during the experiment, initially decreasing and later increasing (Appendix A). As expected from a renal function within a normal range, blood pH remained stable and within normal values throughout the experiment (Appendix A), although the anion gap significantly increased at week 10 (Appendix A). In the absence of overt renal function changes, our results suggest subclinical renal function impairment, as noted by progressive proteinuria, an increase in the urinary protein-to-creatinine ratio, a significant increase in albuminuria at the 10-week time point (Appendix A), and a progressive decline in the BUN-to-creatinine ratio, which still remained within normal values (Appendix A). The remaining blood parameters fluctuated during the experiments but mostly remained within normal values and are shown in Appendix A.

Exercise of either moderate or high intensity did not modify renal function parameters. Although proteinuria was higher in both trained groups at the 10-week time point (Appendix A), both the urinary protein-to-creatinine ratio and albuminuria were similar between groups (Appendix A). In summary, our findings suggest that this animal model recapitulated early nephropathy, and its progression was not modified by exercise at different levels.

### 3.3. Exercise Induces Structural Aortic Remodeling

The aortic root measured in vivo with echocardiography was larger in intensively trained rats than in sedentary rats (Figure 2A). In histological analyses, the collagen deposit in the tunica media was seemingly increased in trained rats, although statistical comparisons were jeopardized by the small sample size of the sedentary obese group (Figure 3A). Similarly, procollagen I and III mRNA expression increased in the two trained groups compared with the sedentary group (Figure 3B,C), while expression of the collagen degradation enzyme matrix metalloproteinase 2 (Mmp2) was similar between obese groups (Figure 3D). The extracellular matrix turnover was remarkably faster in young lean control rats than in aged obese Zucker rats, as demonstrated by a higher expression of procollagens I and III and matrix metalloproteinase 2 (Figure 3D), supporting a senescent phenotype in the aged, obese Zucker rats.

### 3.4. Moderate Exercise Enhances Endothelial Function

Endothelial dysfunction commonly precedes atherosclerosis. We, therefore, assessed vascular reactivity ex vivo in the thoracic aortic rings of all groups. Contraction and relaxation curves are plotted in Figure 4 and Figure 5, and logEC50 and E_max_ estimates and statistical comparisons for all groups and conditions are reported in Table 1. The MOD group showed the best endothelial function among all groups, as demonstrated by a significantly enhanced relaxation response to CCH (higher NO sensitivity, nonsignificant increase in Emax, Figure 4A), a difference that was suppressed by the NO synthase inhibitor LNMMA (Figure 4B). To test whether MOD-improved relaxation under CCH resulted from the enhanced NO sensitivity of vascular smooth muscle cells (VSMCs) in the tunica media, we analyzed endothelium-independent relaxation. Sodium nitroprusside (SNP) was used to elicit the VSMC relaxation response under an endothelial NO synthesis blockade with LNMMA. We found that the VSMCs of MOD rats had a similar NO sensitivity to that of INT rats (Figure 4C), pointing to increased endothelial NO release as the basis for the improved MOD response to CCH. SED rats presented the most sensitive endothelium-independent relaxation, possibly owing to chronic NO deprivation. Finally, the contractile response to phenylephrine was analyzed. All obese rats showed decreased sensitivity to PHE, and among the obese groups, moderately trained rats presented with the lowest logEC50 values (Figure 5A). After adding LNMMA, differences in logEC50 estimates between groups were abolished (Figure 5B). Overall, these results suggest that only moderate training improved endothelial function in this high-CV-risk animal model.

### 3.5. Oxidative Stress Modulates Endothelial Function in Intensively Trained Rats

We next evaluated the potential implication of oxidative stress in endothelial function. A free radical scavenger (TEMPOL) was added to the buffer, and relaxation (CCH) and contraction (PHE) experiments were repeated. Incubation with TEMPOL improved CCH-mediated relaxation in all groups, particularly in the INT group, which became similar, and nonsignificantly different, to MOD rats (Figure 4D and Table 1). After the addition of TEMPOL, PHE-induced contraction became similar in all obese groups (Figure 5C and Table 1).

To further investigate the role of oxidative stress in vascular function, the mRNA expression of oxidative stress-related markers was evaluated in the aortic wall. Compared with lean animals, intensively trained rats presented with an increased expression of pro-oxidant Nox2, while both exercised groups showed an increased expression of P47 (Figure 6A). These findings were accompanied by a higher expression of antioxidant markers (*Cat*, *Sod2*, *Gpx1*, *Nrf2*, and *Gch1*) in the trained groups (Figure 6B). NRF nuclear translocation was analyzed in aortic histological sections by immunofluorescence and yielded no differences between groups (Figure 6C). We next focused on perivascular adipose tissue (PVAT), which has been shown to play an important regulatory role in vascular function. Higher expression of the pro-oxidant marker *Nox2* was found in INT rats (Figure 6D) than in MOD rats. Moreover, INT rats expressed lower levels of the antioxidant markers *Gch1* and *Sod2* (Figure 6E).

### 3.6. Cardiac Phenotype Diverges Depending on Exercise Intensity in Rats at High CV Risk

Among obese rats, the INT group had an increased tibia length-indexed wet heart weight compared to MOD (32.44 ± 0.72 vs. 30.23 ± 0.38, *p*-value = 0.02). Representative long-axis echocardiographic M-mode images placed below the mitral valve leaflets are shown in Figure 2B. Heavily trained rats developed significant eccentric hypertrophy, as demonstrated by a thicker IVS (Figure 2C) and a larger LVDd (Figure 2D). The LVPW was similar in all groups (Figure 2E). Systolic function was estimated through ejection fraction (Figure 2F) and fractional shortening (Figure 2G), which were mildly but significantly reduced in INT rats compared with SED rats. The diastolic function assessed by the E/A ratio was unchanged between groups (Figure 2H).

We subsequently analyzed histological samples to confirm cardiac hypertrophy. Representative LV pictures of triple immunostaining are shown in Figure 7A. LV free wall sections from exercised rats (MOD and INT) showed larger myocytes (Figure 7B), though this difference was only significant for MOD rats in the septum (Figure 7C). To further characterize the cardiac remodeling phenotype underlying eccentric hypertrophy, the mRNA expression of myosin heavy chain (MHC) and titin isoforms was assessed in the IVS. The results for αMHC and βMHC are shown in Figure 8A. The βMHC/αMHC ratio, for which higher values (i.e., βMHC preponderance) denote pathological remodeling, was subsequently calculated (Figure 8B). SED rats showed the highest βMHC/αMHC ratio, while the MOD and INT groups presented a transition toward a lower ratio; lean rats showed the lowest value (Figure 8B). Increased levels of the titin-compliant isoform (N2bA) were expressed by the SED and INT groups compared with lean rats (Figure 8C). Obesity blunted the LV mRNA expression of BNP (Figure 8D), as previously shown [16]. The septum of obese rats expressed lower mRNA levels of the physiological hypertrophy mediator Igfr1 (Figure 8F), while Mef2d, which encodes a transcription factor involved in pathological growth pathways, was upregulated in all obese groups (Figure 8G).

Myocardial collagen deposits were quantified as a hallmark of pathological remodeling. Obese rats showed an increased fibrosis content in the three ventricular walls analyzed (LV, IVS, and RV) compared with that of lean rats; although moderate exercise decreased overall fibrosis, this reduction did not reach statistical significance (FDR-adjusted *p*-value = 0.052, Figure 9A). In triple immunostaining sections, no major differences were found between groups in endomysial fibrosis (Figure 7D,E). Lean rats presented more abundant fibroblast labeling (i.e., vimentin staining) in both the LV free wall and septum than that of obese rats (Figure 7F,G), suggesting faster collagen and ECM turnover in younger, healthy rats. In keeping with these results, the mRNA expression of procollagen types I and III and matrix metalloproteinase 2 was higher in control rats than in obese rats (Figure 9B–E). These findings are consistent with a senescent phenotype in high-CV-risk rats, which is not significantly modified by any form of exercise.

### 3.7. Intensive Exercise Tends to Induce an Inflammatory Profile

Among obese rats, those who performed intensive training showed higher plasma levels of TNFα at rest (Appendix A). Conversely, IL-6, IL-10, adiponectin, sE-selectin, and sI-CAM1 levels were comparable among the groups (Appendix A–F).

## 4. Discussion

In the present study, we aimed to assess the effect of two different loads of exercise in a high-cardiovascular-risk animal model recapitulating obesity, metabolic syndrome, aging, and a mildly impaired glomerular filtration rate. Our results show that (1) both exercise loads decreased the rats’ overweight status and improved the metabolic profile; (2) moderate but not intense exercise improved vascular endothelial function; (3) intense exercise prompted a pro-oxidant profile in aortic and perivascular aortic adipose tissue that might substantiate the lack of endothelial function improvement; (4) moderate exercise had the most intense obesity-induced cardiac remodeling preventive effect; and (5) neither moderate nor intense exercise impinged the progression of mild forms of reduced glomerular filtration reduction. These results support moderate-intensity exercise as the optimal and most effective form of exercise to prevent cardiovascular disease even for individuals at high risk.

### 4.1. An Animal Model to Fill the Gap of Very Intense Exercise in Very High-Risk Patients

The net outcome of physical activity on CV health relies on the balance between its direct and indirect beneficial effects on the one hand and potential induced harm on the other hand. Coronary calcification has been demonstrated in master marathon runners free of risk factors [6,7,17], but exercise has been shown to decrease ischemic risk in aged individuals or those accruing several risk factors, independent of intensity. Based on the latter, current policies recommend as much exercise as possible in patients at high-CV risk. Nevertheless, there is a paucity of data to support the limits of this claim. These individuals usually perform moderate to vigorous exercise, and the number of individuals at this extreme, even in large epidemiological studies, is low and highly selected.

Animal models have evolved as a unique opportunity to address exercise at the extremes in an unbiased way, but rodents are known to be resistant to atherosclerosis. Transgenic mice have been classically used in this setting, but their use is focused on single mechanisms and pathways and may fail to depict the complex interplay of risk factors in the development of atherosclerosis. We, therefore, developed a high-risk animal model in the absence of life-threatening conditions (e.g., severe renal failure) or genetic manipulation to test clinically relevant hypotheses. The obese Zucker model accrues several CV risk factors, including obesity, hypertension, and insulin resistance. Moreover, aged male rats were selected to yield a higher risk and more severe metabolic syndrome. Metabolic syndrome is commonly associated with glomerulosclerosis and induces renal failure, while Zucker rats only show early stages of renal disease, including hyperfiltration [18]; therefore, a single nephrectomy was performed to impair renal function. As a result, this high-risk animal model recapitulated several CV complications, including pathologic myocardial remodeling and endothelial dysfunction, and appears to be a suitable model to study the effects of strenuous exercise training in this substrate.

### 4.2. Improved Risk Profile in Heavily Trained Rats Is Not Paralleled by Vascular Prevention

At least partially, the beneficial effects of exercise are mediated by mitigating the severity of CV risk factors. In our model, moderate or very high-intensity exercise improved the overall CV risk profile by blunting obesity and improving glycemic metabolism. Glucose metabolism abnormalities have been linked with the augmented CV risk observed in obesity, largely mediated by endothelial dysfunction [19]. The benefits of exercise training on insulin signaling have been emphasized by experimental [20] and clinical studies showing a 60% reduction in diabetes diagnosis as a result of regular exercise in individuals at high risk [21].

We consequently expected that metabolic improvement would translate into a lower risk of atherosclerosis. Although atherosclerotic plaques were absent, endothelial dysfunction served as a surrogate marker [22]. Sedentary rats presented with impaired endothelial function, as evidenced by a reduced endothelial-mediated NO release and relaxation response. While moderate exercise significantly improved endothelial dysfunction, strenuous exercise failed to show the same effects. A blockade of NO synthesis essentially equaled the reactivity of all groups, underscoring that the endothelium is central in this process.

Interestingly, in our model, PHE-induced contraction decreased in all high-risk groups compared with 10-week-old lean rats, likely due to a senescent vascular phenotype in aged, 36-week-old rats accruing several risk factors. Aging is associated with a decrease in the number of phenylephrine receptors in aortic VSMCs [23], leading to a decreased vascular contractile response [23,24]. Similarly, insulin resistance impinges endothelial function and VSMCs and promotes a blunted response to PHE [25]. Thereby, both senescence and obesity may substantiate a lower responsiveness to PHE in the high-cardiovascular-risk group.

These results are similar, but do not overlap, with previous work conducted on healthy rats [9]. In both healthy and high-risk rats, moderate exercise significantly improved endothelial function. Conversely, outcomes differed for strenuous exercise: in healthy animals, endothelial NO release peaked in intensively trained rats, but this was partially blunted by vasoconstricting agents, eventually resulting in similar endothelial function after moderate and strenuous exercise [9]. Here, we show that endothelial function did not improve in high-risk animals after strenuous exercise, denoting that the most intense vascular affectation cannot be reversed by strenuous exercise. In keeping with our findings, the endothelial function of hypertensive rats did not benefit from strenuous exercise in previous work [26]. These findings highlight that the consequences of physical activity rely on its intensity but also the baseline risk and uncover the complex relationship between exercise and cardiovascular risk.

Interestingly, our data implicate oxidative stress and inflammation as central contributors to explain the lack of improved endothelial function in heavily trained rats. In intensively trained rats, we found that a free radical scavenger matched the endothelial function to that of rats trained at a moderate pace, along with a pro-inflammatory status in the PVAT and an increase in plasma TNF-α. Notably, the role of oxidative stress in PVAT is increasingly being recognized as regulating endothelial dysfunction and contractile regulation [27,28]. A pro-inflammatory environment is characteristic of individuals at a high risk of vascular diseases [29]; strenuous bouts of physical activity are associated with transient, intensity-dependent increases in proinflammatory markers [30]. It is then plausible that risk factor- and strenuous exercise-induced inflammation and oxidative stress may synergistically combine to offset the metabolic syndrome mitigation promoted by exercise. Additional research is needed to confirm this hypothesis.

### 4.3. Moderate Intensity Exercise Yields Larger Benefits in Preventing Obesity-Induced Cardiac Remodeling

Both obesity and physical activity are strong triggers for structural and functional cardiac remodeling. On the one hand, obesity elicits LV hypertrophy, diastolic dysfunction, and myocardial fibrosis, which may evolve into systolic dysfunction in some cases [31,32]. Our high-cardiovascular-risk model recapitulated most of these findings, including myocardial hypertrophy and fibrosis, which were accompanied by a transition toward an overexpression of βMHC and a relative predominance of N2ba. αMHC is the most abundant myosin heavy chain isoform, whereas a shift toward the expression of βMHC occurs in heart failure and pressure overload-induced hypertrophy [33], which may mediate a progression toward pathological remodeling and dysfunction [34]. The balance between titin N2b and N2ba isoforms determines myocardial compliance [35], in which a relative increase in N2ba is associated with a stiffer LV and diastolic dysfunction [36].

On the other hand, exercise has been shown to promote physiological cardiac remodeling known as “athlete’s heart”, which enhances cardiac efficiency and its ability to increase output. Moderate exercise has been demonstrated to blunt the deleterious structural remodeling induced by obesity; our results are in line with these findings. Nevertheless, the effect of high-intensity training on obesity has not yet been defined. In our model, we found that high-intensity training promotes eccentric hypertrophy, similar to the athlete’s heart pattern. However, the relevance of a mild reduction in LV ejection fraction in heavily trained rats remains obscure. Such a reduction is commonly regarded as physiological and, along with eccentric hypertrophy, a regular component of the athlete’s heart spectrum which results in the maintenance of stroke volume. These effects would serve as a volume reservoir to accommodate the large increases in cardiac output during exercise bouts. A regression of the obesity-induced increase in the β- to α-MHC ratio supports this hypothesis [37,38]. Intriguingly, however, and in contrast to moderate exercise, intense exercise could not reverse the obesity-induced titin isoform imbalance (Figure 7D) or blunt myocardial fibrosis (Figure 9A).

Our results suggest that while no deleterious effect of high-intensity exercise on LV structural and functional remodeling was evident, moderate exercise elicited a larger benefit. The LV has commonly been protected from the deleterious cardiovascular effects of high-intensity exercise [39,40], possibly due to its superior ability to accommodate pressure and volume overload during exercise [11]. Few reports have shown an increased fibrosis burden in the LV, but its origin is uncertain and related to increased asymptomatic myocarditis or coronary artery disease [41].

### 4.4. Clinical Implications

A U-shaped relationship between exercise load and disease incidence has been robustly established in healthy individuals for AF [5] and suggested for coronary disease [6,9,10]. Our data support that a similar relationship could be established for individuals at high CV risk, in whom moderate-intensity physical activity would also yield the maximal improvement in CV physiology, i.e., benefits are blunted when exercise turns strenuous. Clinical data support our findings. In patients with stable coronary disease, weekly hours of strenuous exercise are a strong predictor of mortality; a lower mortality rate is reached at 10 to 11 h per week, subsequently increasing in parallel with frequency [42]. Moreover, in myocardial infarction survivors, engaging in exercise for >7.2 MET-h/day was associated with increased mortality compared with 7.2 MET-h/day (HR 2.62, *p* = 0.009) [43]. Notably, patients at a very high risk seldom perform long-term high-intensity exercise, and our data encourage these patients to refrain from strenuous exercise. Unfortunately, our work may not be useful to establish a safety/danger threshold. Translation from animal models to humans should always be considered carefully. Correlations between rodents and humans in terms of their lifespan and physiological adaptation to different degrees of exercise intensity and duration may only be considered rough estimates. The animal model reported here is to be interpreted as long-term, approximately 10 years in human life [9]. In addition, intensity may only be very roughly approximated, ranging from 50 to 60% of VO_2max_ in the MOD group to 85% in the INT group. Specifically, dedicated studies in humans are urgently warranted.

In summary, our results support the concept that the maximal benefits of physical activity are reached when performed at a moderate intensity even for individuals with several CV risk factors. Current clinical and translational evidence supports that moderate to vigorous exercise should be recommended for individuals at high CV risk, and we believe that individuals should be warned that strenuous exercise provides no additional health benefits or, in some cases, has even proved deleterious. We acknowledge, though, that additional incentives and rewards, beyond health, may derive from strenuous exercise.

### 4.5. Limitations

Some limitations of our work should be acknowledged. First, rodents are known to be atherosclerotic resistant, and, therefore, only early vascular disease markers were assessed. Experiments in atherosclerosis-prone transgenic animals may provide additional insights. Second, we aimed to test the effects of exercise in animals at the highest cardiovascular risk, and thereby we enriched them with several CV risk factors, including male sex, aging, glucose intolerance, and a reduced glomerular filtrate rate. Caution needs to be taken when extrapolating our findings to the female sex or rats accruing other risk factors. Moreover, alternative types of exercise, such as strength or interval training, may yield different outcomes. Finally, we did not perform an a priori formal sample size calculation, and thereby the statistical power of our analyses, particularly those showing no significant differences, is uncertain.

## 5. Conclusions

In this animal model accruing several CV risk factors, including senescence, metabolic syndrome, and a mild reduction in glomerular filtration, we found impaired endothelial function and deleterious cardiac remodeling. Long-term moderate exercise partially blunted the pathological phenotype; however, strenuous training could not regress endothelial dysfunction and yielded a lower improvement in LV remodeling. Oxidative stress and inflammation may be responsible for this differential effect of strenuous exercise. Our results underline the central role of moderate, but not strenuous, exercise in the management of CV risk factors.

## Figures and Tables

**Figure 1 antioxidants-12-01462-f001:**
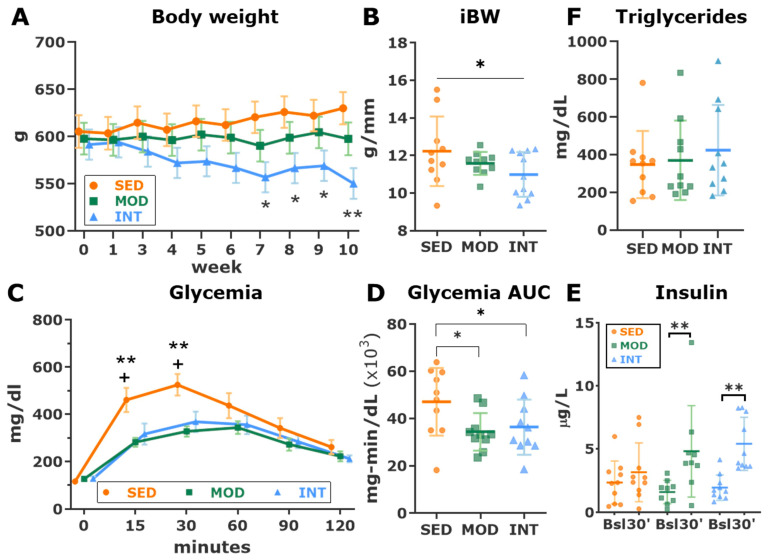
Impact of different loads of regular training on cardiovascular risk factors. (**A**) Weekly measurement (estimated marginal mean ± standard error of the mean {SEM}) of body weight. Analysis with linear mixed modeling with a repeated measures covariance structure, including week, group, and its interaction as predictors. The time point × group interaction was significant at the *p* < 0.001 level; within each week, significant post hoc pairwise comparisons between groups (LSD tests) are shown. N = 10 per group. * *p* < 0.05, ** *p* < 0.01 INT vs. Sed (**B**) Indexed body weight (BW) before euthanasia. Analysis with a linear trend analysis. * *p* < 0.05. (**C**) Results (estimated marginal mean ± SEM) of the intraperitoneal glucose tolerance test. Analysis with linear mixed modeling with a repeated measures covariance structure, including time point, group, and its interaction as predictors. The time point × group interaction was significant at the *p* < 0.001 level; within each time point, significant post hoc pairwise comparisons between groups (LSD tests) are shown. N = 10 per group. ** *p* < 0.01 SED vs MOD; + *p* < 0.05 SED vs INT. (**D**) Area under the curve (AUC, mean ± SD) of glucose blood concentrations during the glucose test. Analysis was performed with a one-way ANOVA which showed a significant omnibus test at the *p* < 0.05 level. Significant pairwise LSD tests are shown. * *p* < 0.05. (**E**) Serum insulin concentration (mean ± SD) before and 30 min after glucose administration. Analysis with linear mixed modeling with a repeated measures covariance structure, including time point (i.e., baseline and 30 min) and group and its interaction as predictors. The interaction time point × group was significant at the *p* < 0.05 level; within each group, significant post hoc pairwise comparisons between time points (baseline vs. 30-min, LSD tests) are shown. ** *p* < 0.01. (**F**) Triglyceride plasma concentration (mean ± SD) in all groups. Analysis was performed with a one-way ANOVA. SED, MOD, and INT (obese sedentary, moderately and intensively trained rats, respectively).

**Figure 2 antioxidants-12-01462-f002:**
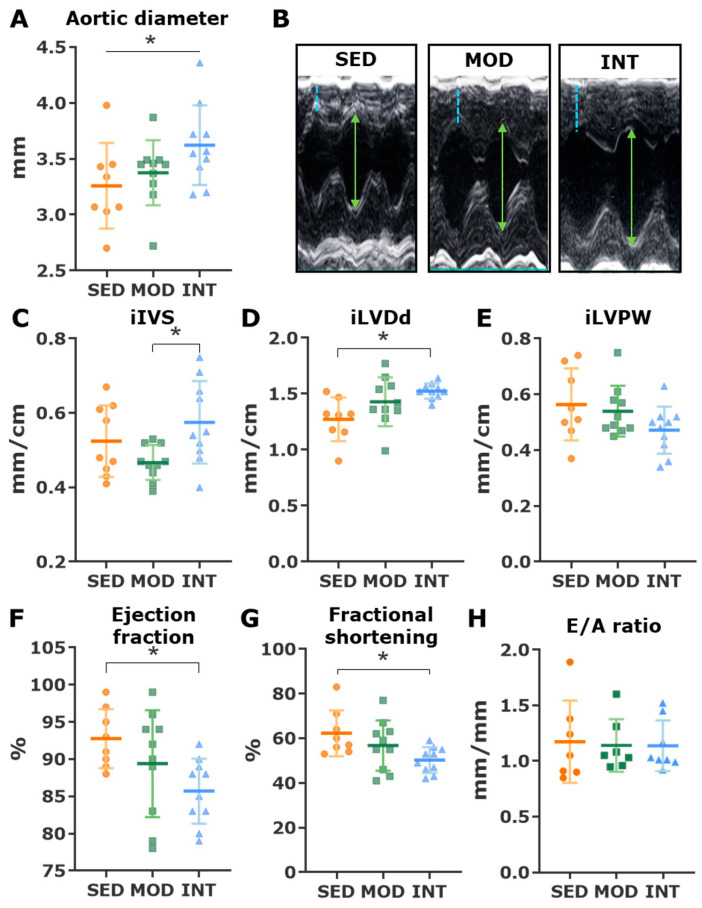
Echocardiographic changes produced by exercise. (**A**) Ascending aortic diameter (mean ± SD). Analysis with a linear trend analysis. (**B**) Representative M-mode images of a parasternal long-axis view. The segmented blue line identifies the interventricular septum (IVS) and the green arrow identifies the left ventricular diastolic diameters (LVDd). (**C**–**H**) The mean ± SD is shown for indexed IVS (iIVS) (**C**), indexed LVDd (iLVDd) (**D**), indexed left ventricle posterior wall (iLVPW) (**E**), left ventricle ejection fraction (**F**) and fractional shortening (**G**), and transmitral E/A ratio (**H**). All analyses were performed with a one-way ANOVA which showed a significant omnibus test at the *p* < 0.05 level for iVS, iLVDd, ejection fraction, and fractional shortening. Post hoc pairwise comparisons were performed with the LSD tests. * *p* < 0.05. SED, MOD, and INT (obese sedentary, moderately and intensively trained rats, respectively).

**Figure 3 antioxidants-12-01462-f003:**
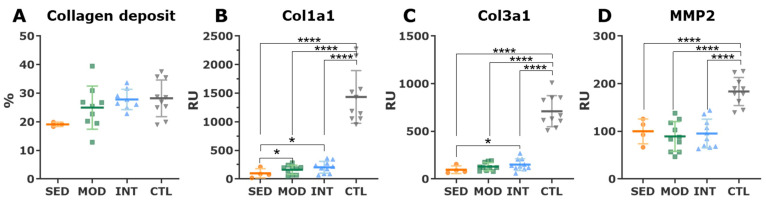
Aortic collagen deposition and turnover. (**A**) Percentage of collagen deposit in the tunica media from picrosirius red-stained thoracic aorta (mean ± SD). (**B**–**D**) Aortic mRNA expression (mean ± SD) of alpha-1 procollagen types 1 (**B**) and 3 (**C**), and matrix metalloproteinase-2. Analyses were performed with one-way ANOVA which showed a significant omnibus test at the *p* < 0.0001 level for gene expression parameters. Post hoc pairwise comparisons (*t*-tests) were FDR-adjusted. * *p* < 0.05; **** *p* < 0.0001. SED, MOD, and INT (obese sedentary, moderately, and intensively trained rats, respectively), CTL (young lean rats).

**Figure 4 antioxidants-12-01462-f004:**
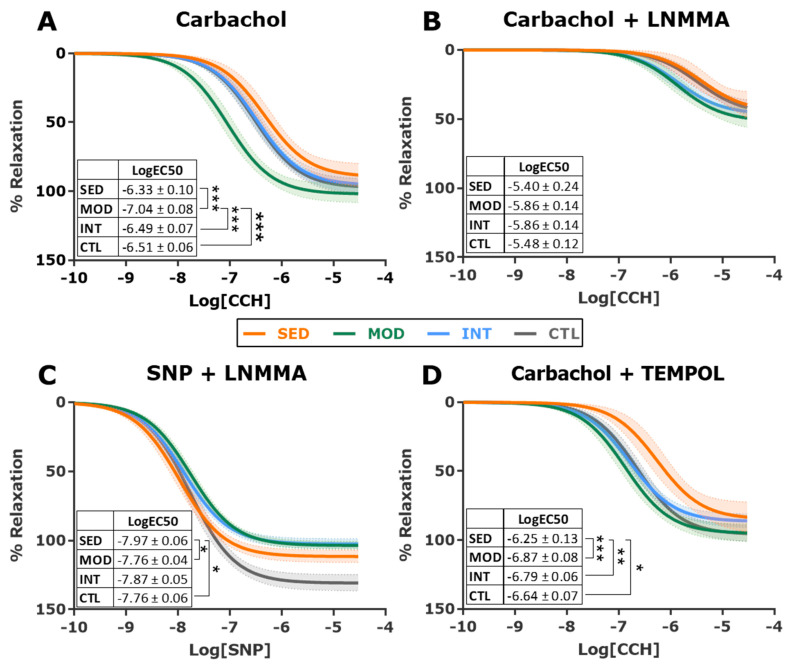
Relaxation response in vascular reactivity experiments in descending thoracic aorta. Dose-response relaxation curves induced by carbachol (CCH) alone (**A**) and in the presence of nitric oxide inhibitor LNMMA (**B**), sodium nitroprusside (SNP) in the presence of L-NMMA (**C**), and carbachol in the presence of the free radical scavenger TEMPOL (**D**). Shaded areas of all curves represent 95% CI. The estimated logEC50 (mean ± SEM) for each group and curve is shown in the inset. Analyses were performed with a one-way ANOVA per each individual graph, comparing fitted logEC50 in three parameter equations. Significant post hoc pairwise FDR-adjusted comparisons between groups are shown, * *p* < 0.05, ** *p* < 0.01, and *** *p* < 0.001. SED, MOD, and INT (obese sedentary, moderately, and intensively trained rats, respectively), CTL (young lean rats). For all experiments: n = 9 (SED), n = 10 (MOD), n = 10 (INT, except for panel 4C in which n = 9), and n = 8 (CTL).

**Figure 5 antioxidants-12-01462-f005:**
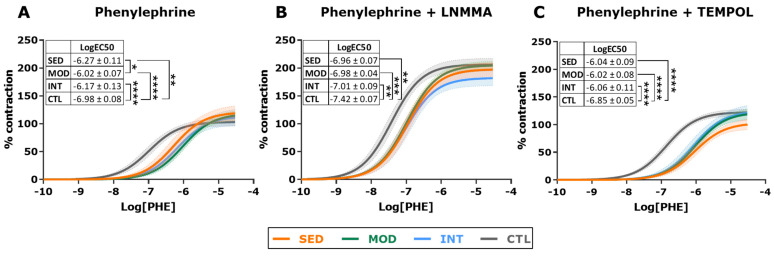
Contractile response in vascular reactivity experiments in descending thoracic aorta. Dose-response contraction curves induced by phenylephrine (PHE) alone (**A**) and in the presence of the nitric oxide inhibitor L-NMMA (**B**) or the free radical scavenger TEMPOL (**C**). Shaded areas of all curves represent 95% CI. Estimated LogEC50 (mean ± SEM) for each group and curve is shown. Analyses were performed with a one-way ANOVA per each individual graph, comparing fitted logEC50 in three parameter equations. Significant post hoc pairwise FDR-adjusted comparisons between groups are shown, * *p* < 0.05, ** *p* < 0.01, *** *p* < 0.001, and **** *p* < 0.0001. SED, MOD, and INT (obese sedentary, moderately, and intensively trained rats, respectively), CTL (young lean rats). For all experiments: n = 6 (SED, except for panel (**C**) in which n = 5), n = 11 (MOD), n = 9 (INT), and n = 8 (CTL).

**Figure 6 antioxidants-12-01462-f006:**
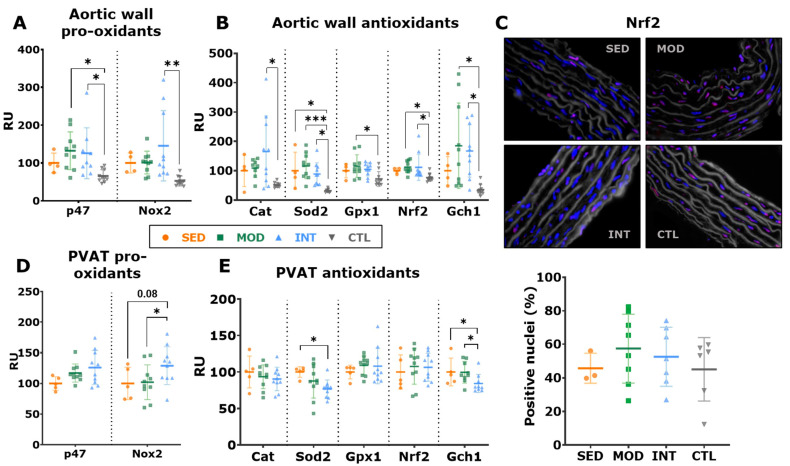
Oxidative stress imbalance in aortic and perivascular adipose tissue. (**A**) Aortic mRNA expression (mean ± SD) of pro-oxidant markers. (**B**) Aortic mRNA expression (mean ± SD) of antioxidant markers. (**C**) Representative images of Nrf2 nuclei translocation in the aortic thoracic wall (upper panel; nuclei stained in blue {DAPI}, Nrf2 in red, when translocated to the nuclei appears pink/purple), and percentage (mean ± SD) of positive nuclei in the tunica media (lower panel). (**D**) Perivascular adipose tissue (PVAT) mRNA expression of pro-oxidant markers (mean ± SD). (**E**) PVAT mRNA expression of antioxidant markers (mean ± SD). Analyses were performed at the gene level with a one-way ANOVA, significant post hoc pairwise comparisons between groups (FDR adjustment for (**A**,**B**), LSD tests for (**D**,**E**) are shown, * *p* < 0.05, ** *p* < 0.01, *** *p* < 0.001. SED, MOD, and INT (obese sedentary, moderately and intensively trained rats, respectively), CTL (young lean rats).

**Figure 7 antioxidants-12-01462-f007:**
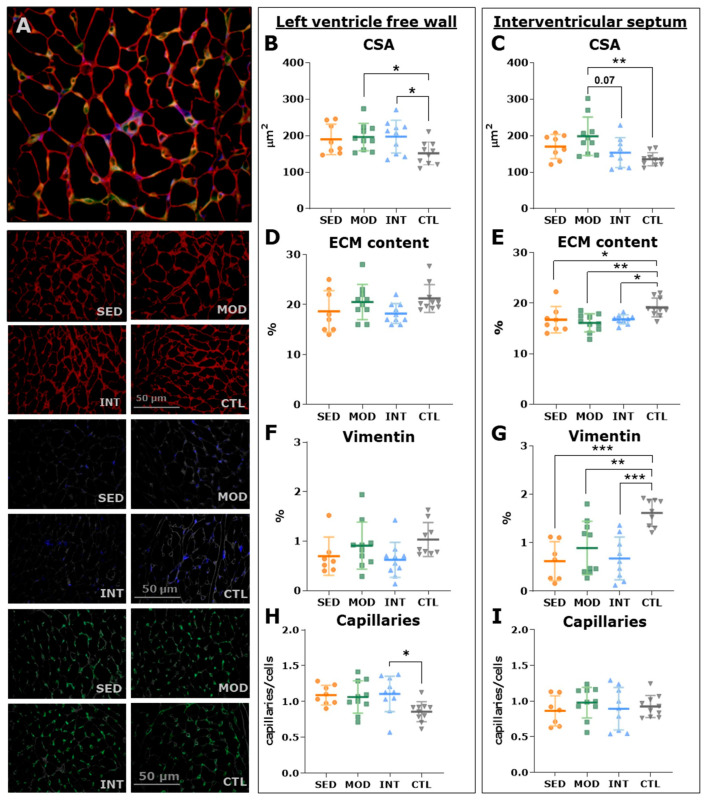
Left ventricular myocardial histological assessment. (**A**) Representative triple immunofluorescence images of myocardial triple immunofluorescence (wheat germ agglutinin in red for extracellular matrix, vimentin in blue for fibroblasts, and isolectin-GS IB4 in green for capillaries). Results (mean ± SD) for the left ventricular (LV) myocyte cross-sectional area (CSA) in the LV free wall and the interventricular septum (IVS) ((**B**,**C**), respectively), extracellular matrix content in the LV free wall and the IVS (**D**,**E**), vimentin-positive area in the LV free wall and the IVS (**F**,**G**), and capillary density in the LV free wall and the IVS (**H**,**I**). Analyses were performed with a one-way ANOVA, and significant post hoc pairwise FDR-adjusted comparisons between groups are shown, * *p* < 0.05, ** *p* < 0.01, and *** *p* < 0.001. SED, MOD, and INT (obese sedentary, moderately, and intensively trained rats, respectively), CTL (young lean rats).

**Figure 8 antioxidants-12-01462-f008:**
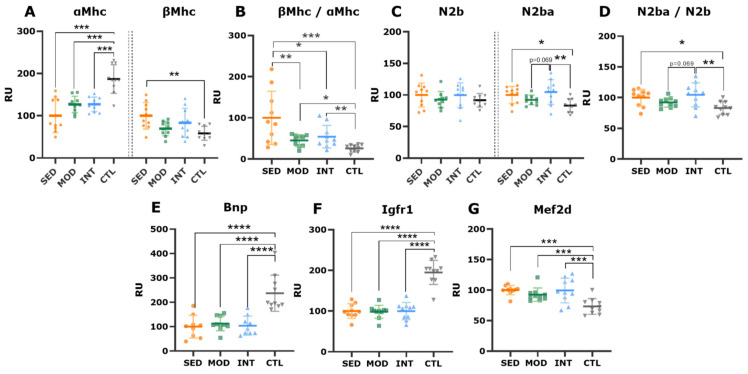
Expression of genes involved in cardiac remodeling in all groups. Interventricular septum mRNA expression (mean ± SD) of the α- and β-myosin heavy chain isoforms (Mhc) (**A**) and their ratio (**B**), titin isoforms N2b and N2ba (**C**), and their ratio (**D**), Bnp (**E**), Igfr1 (**F**), and Mef2d (**G**). Analyses were performed at the gene level with a one-way ANOVA, significant post hoc pairwise FDR-adjusted comparisons between groups are shown, * *p* < 0.05, ** *p* < 0.01, *** *p* < 0.001, and **** *p* < 0.0001. SED, MOD, and INT (obese sedentary, moderately, and intensively trained rats, respectively), CTL (young lean rats).

**Figure 9 antioxidants-12-01462-f009:**
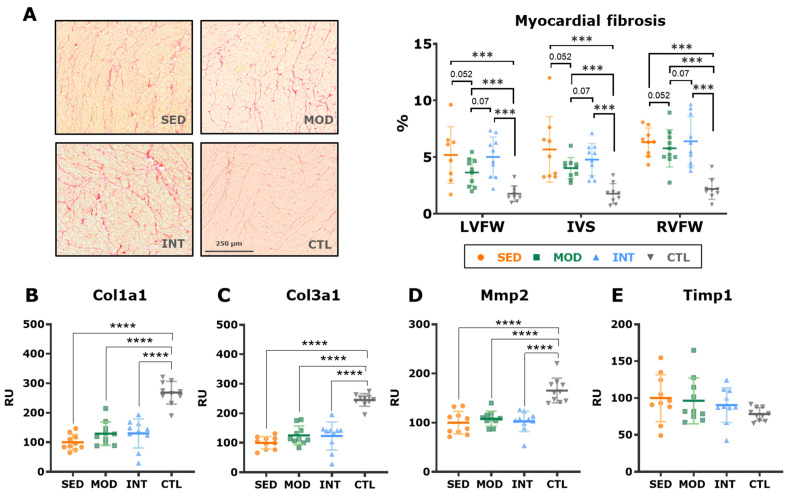
Myocardial collagen deposition. (**A**) Representative images of picrosirius red-stained samples (**left panel**) and quantification of myocardial fibrosis (mean ± SD, **right panel**) in the left ventricle free wall (LVFW), interventricular septum (IVS), and right ventricle free wall (RVFW). Analysis was performed with a linear mixed effects modeling, including group, chamber, and their interaction as predictors; both group and cardiac chamber were significant at the *p* < 0.001 level, and significant post hoc pairwise comparisons within the group main factor are shown (*p*-values are FDR-adjusted). (**B**–**E**) mRNA expression (mean ± SD) of alpha-1 procollagen types 1 (**B**) and 3 (**C**), matrix metalloproteinase-2 (**D**), and tissular metalloproteinase inhibitor 1 (**E**). Analyses were performed at the gene level with a one-way ANOVA, and significant post hoc pairwise FDR-adjusted comparisons between groups are shown. *** *p* < 0.001, and **** *p* < 0.0001. SED, MOD, and INT (obese sedentary, moderately, and intensively trained rats, respectively), CTL (young lean rats).

**Table 1 antioxidants-12-01462-t001:** Vascular reactivity estimates (mean ± SEM) of all groups and conditions.

	LogEC50					Emax				
	SED	MOD	INT	CTL	ANOVA (Omnibus p)	SED	MOD	INT	CTL	ANOVA (Omnibus p)
CCH	−6.33 ± 0.10 ^###^	−7.04 ± 0.08 ***	−6.49 ± 0.07 ***	−6.51 ± 0.06	<0.0001	−89.54 ± 4.43	−101.98 ± 3.36	−95.67 ± 3.49	−97.48 ± 2.99	0.13
CCH + LNMMA	−5.40 ± 0.24	−5.86 ± 0.14	−5.86 ± 0.14	−5.48 ± 0.12	0.11	−44.91 ± 7.57	−51.43 ± 4.15	−46.47 ± 3.74	−46.22 ± 3.99	0.33
SNP + LNMMA	−7.97 ± 0.06 *^,#^	−7.76 ± 0.04	−7.87 ± 0.05	−7.76 ± 0.06	0.02	−111.54 ± 2.27 *	−103.62 ± 1.46 *	−102.51 ± 1.81 *	−130.77 ± 2.94	<0.0001
CCH + TEMPOL	−6.25 ± 0.13 *^,###,&^	−6.87 ± 0.08	−6.79 ± 0.06 ^&^	−6.64 ± 0.07	0.0003	−84.83 ± 5.82	−95.03 ± 3.21	−86.48 ± 2.53	−95.89 ± 3.06	0.10
PHE	−6.27 ± 0.11 **^,#^	−6.02 ± 0.07 ***	−6.17 ± 0.13 ***	−6.98 ± 0.08	<0.0001	120.90 ± 7.01	118.39 ± 4.77	113.65 ± 7.93	103.39 ± 3.59	0.26
PHE + LNMMA	−6.96 ± 0.07 **	−6.98 ± 0.04 ***	−7.01 ± 0.09 **	−7.42 ± 0.07	0.0005	197.92 ± 6.31	205.10 ± 3.92	182.37 ± 6.88 *^,#^	206.53 ± 5.60	0.01
PHE + TEMPOL	−6.04 ± 0.09 ***	−6.02 ± 0.08 ***	−6.06 ± 0.11 ***	−6.85 ± 0.05	<0.0001	103.24 ± 5.16	122.47 ± 5.28	124.63 ± 7.51	122.65 ± 2.96	0.14

LogEC50 and Emax estimates for vascular reactivity experiments. Curves are plotted in Figure 4 and Figure 5. * *p* < 0.05, ** *p* < 0.01, *** *p* < 0.001 vs. CTL; ^#^*p* < 0.05, ^###^*p* < 0.001 vs. MOD; and ^&^
*p* < 0.05 vs. INT. Post hoc pairwise comparisons were FDR-adjusted.

## Data Availability

Full data are available from the corresponding authors upon reasonable request.

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
