# Peer review of "High-Intensity Exercise Promotes Deleterious Cardiovascular Remodeling in a High-Cardiovascular-Risk Model: A Role for Oxidative Stress"

_antioxidants, 2023, doi:10.3390/antiox12071462_

Round 1

Reviewer 1 Report

Dear authors,

If is possible add some statisctical data in abstract.

Try to add in the introduction more information about high intensity exercise and oxidative stress.

The manuscript is well designed. Congratulations.

Reviewer 2 Report

Dear authors

Your research is very interesting, however I could say that I would have doubts as to whether the results of this research in an even unknown animal sample can be generalized to humans. In addition to obesity, there are other factors that can affect the outcome of intense training in high-risk patients, such as age (here the experimental animals were relatively young, psychological factors, coexistence of smoking, etc.)

Please try to justify how your results can be generalized to human and what it the practicality/usefuleness in clinical practice.

Try to improve the introduction, aim should be clearer. Discussion whould be improved about correlations between rats and human.

Recommended REFs for Intro/Discussion

Effects of Exercise and L-arginine on Ventricular Remodeling and Oxidative Stress, doi: 10.1249/MSS.0b013e3181b2e899

Effects of high intensity interval training on pregnant rats, and the placenta, heart and liver of their fetuses. https://doi.org/10.1371/journal.pone.0143095

Line 74-82:

Tele-assessment of Functional Capacity through the Six-Minute Walk Test in Patients with Diabetes Mellitus Type 2: Validity and Reliability of Repeated Measurements. Sensors, 23(3), 1354. https://doi.org/10.3390/s23031354

Reviewer 3 Report

The manuscript deals with an interesting scientific question. I congratulate the authors to their tremendous work. I have some methodological remarks:

1) Pretest for normal distribution are not recommended and should not be done. It is more important to check model assumption for the ANOVA / LMM models by visually checking QQplots of residuals and residuals vs fitted plots regarding homoscedasticity.

2) One-way ANOVA is not really necessary as an omnibus test if pair-wise comparisons are done anyway. Then it is enough to perform the pair-wise tests directly (and maybe adjusting for multiplicity). A rationale for multiplicity adjustment if performed should be given. LSD test as post hoc test is not really recommended if the authors want to adjust for multiplicity bc it has only weak FWER control, Tukeys test or FDR based tests are recommended.

3) Relating to point 2) one recommendation: if the number of groups is k=3 then no further multiplicity adjustment of the FWER is necessary if you perform the ANOVA as gatekeeping omnibus test anyway. You can use unadjusted t-tests afterwards.

4) Why were only male rats used in the experiment? It is recommeded to use both sexes in experimental designs or give a really convincing rationale why this was not done. You cannot exclude sex-specific effects that should be investigated as well. CVD due to obesity etc. and different physical activity levels is not limited to male patients only in the clinical setting. This is a huge weakness and limitation of the study. Please inticate the foundation of this rationale.

5) Result figures are missing information on the sample size per treatment group and timepoint etc. Please improve that. How was the appropriate sample size determined in advance? Was there a sample size calculation done in advance?

6) Fig 1 legend E: there is a mistake in the discription: The interaction timepoint x group was significant ...."

7) What is the reason for the differing sample sizes over the different results plots? This should be explained and is certainly a limitation of the study if there were drop outs on the sample for several parameters.

8) Figure legends miss the information which pair-wise post hoc tests were applied to obtain p-values and if any multiplicity adjustment was performed. I my view multiplicity adjustment would not be necessary as this study is more of a hypothesis generating study but if it was performed it may strengthen the results as findings might be more robust.

9) I could not check the supplementary material as the download did not work.

Reviewer 4 Report

This manuscript is well written and has scientific soundness and practical application in every-day life. Moreover, it can be translated into human research which might have high relevance.

However, authors must improve some features such as

-         At the ‘Animal model and exercise protocol’ section (point 2.1 in Material and Methods)

o   Report the number of animals used in each group in the Material and Methods section, point 2.1 Animal model and exercise protocol

o   Why have authors used a younger group of animals for the sedentary group; these animals should be similar to the one’s in the active group in order to control bias associated with confounders such as age, obesity, cardiovascular risk, …

-         Section 2.4.

o   Define LVVd and LVVs used in formulas at the end of section 2.4; I understand the meaning but it should be explicitly defined

o   Numerate these formulas

-         At the ‘statistical analysis’ section (point 2.9)

o   Have you used the Lillefors correction in Kolmogorov-Smirnov test? If you used it, please refer to it; if you didn´t use it, you should have applied it (alternatively you may always apply the Shapiro-Wilk test as its power is never worse than the power of the first one)

o   It is preferable to present SD than SEM; I understand that error bars are smaller but it is not so correct. Given the groups' sample size (which you do not mention) it is possible to compute the SEM.

o   It seems forced to apply a mixed model with a small sample as yours. You do not mention sample size but by counting the points in plots (which are jittered and I assume that they are not overlapped). Authors should explore other assumptions of mixed model analysis and compute achieved power on their experiments (the ones with statistically significant results)  

o   In repeated measures ANOVA it is very important to test for sphericity, more than the normality of the residuals. Have you addressed this assumption? It was not reported.

o   I recognize GraphPad plots but I do not identify where have you used the R Language. If you used any package previously defined by others it should be referenced.

-         ‘Results’ section

o   Always report the p-values with three significant digits and avoid reporting p < 0.05 or p< something. This type of reporting reduces reproducibility in science and also invalidates the eventual use of your paper in a potential systematic review with meta-analysis.

o   Plots such as B, D, F, E in figure 1 (and correspondently in all the figures) could be presented with the points, the middle horizontal line for the mean and limits of confidence, without plotting the bars. I understand that those plots have been being used for too long but they have been incorrectly used. Bars should denote counts (or percentages of counts) and never summary of quantitative data. Please remove the bars and leave just the points and SD (not SEM) in all those figures

o   Instead of * or ** or + you may write the p-value found with 3 significant digits (if significance is below that, just report p < 0.001, you do not need to go below that level)

-         ‘Discussion’ section (point 4.5, Limitations)

o   Some considerations should be addressed for the sample size, assumptions of statistical tests (namely repeated measures ANOVA and linear mixed models), and their impact on the statistical power (which must be computed, a posteriori, if sample size was not computed a priori)  

The quality of the english writing is good

Round 2

Reviewer 2 Report

Μost of the comments were addressed.

Minor editing of English language required
